# Global and Regional Snow Cover Decline: 2000–2022

Stephen S. Young

Geography and Sustainability Department, Salem State University, Salem, MA 01970, USA;
syoung@salemstate.edu

**Abstract:** Snow cover affects the global surface energy balance and, with its high albedo, exerts a cooling effect on the Earth's climate. Decreases in snow cover alter the flow of solar energy from being reflected away from Earth to being absorbed, increasing the Earth's surface temperature. To gain a global understanding of snow cover change, in situ measurements are too few and far between, so remotely sensed data are needed. This research used the medium-resolution sensor MODIS on the Terra satellite, which has been observing global snow cover almost daily since the year 2000. Here, the MOD10C2 eight-day maximum value composite time series data from February 2000 to March 2023 were analyzed to detect global and regional trends in snow cover extent for the first 23 years of the 21st century. Trends in snow cover change during different time periods (seasons and snow-year) were examined using the Mann—Kendall test and the univariate differencing analysis. Both methods produced similar results. Globally, snow cover declined two to ten times as much as it increased, depending on the season of analysis, and annually, global snow cover decreased 5.12% (not including Antarctica or Greenland) based on the Mann—Kendall test at the 95th percentile ($p < 0.05$). Regionally, Asia had the greatest net area decline in snow cover, followed by Europe. Although North America has the second-largest extent of snow cover, it had the least amount of net decreasing snow cover relative to its size. South America had the greatest local decline in snow cover, decreasing 20.60% of its annual (snow-year) snow cover area. The Australia–New Zealand region, with just 0.34% of the global snow cover, was the only region to have a net increase in snow cover, increasing 3.61% of its annual snow cover area.

**Keywords:** snow cover extent; MODIS; Mann—Kendall test; univariate differencing

## 1. Introduction

Snow and ice influences the global surface energy balance and hydrologic cycle as well as modifying feedbacks that control these aspects of the world's climate [1]. When snow cover melts, less solar radiation reflects to space and more of this energy is absorbed by the Earth, and this snow cover feedback is a probable contributor to the polar amplification, which is further warming Arctic regions [2–5]. Various studies have explored the relationship between snow and ice cover, Earth's albedo, and surface energy [6,7], showing that when snow cover declines, albedo decreases and the Earth's surface warms [8–10]. Snow also provides a critical short-term water storage mechanism for many people around the world [11]. Snow cover extent (SCE) has been declining in many parts of the world [8,12,13]. These observed changes in SCE are a direct response to climate change, mainly from warming temperatures and changes in precipitation [14–17]. Climate projections indicate that in the future, SCE will continue shrinking [14], enhancing the snow cover feedback and leading to warmer temperatures [12,15]. Even though SCE globally has decreased in the last four decades, there has been considerable inter-annual variability [18]. Anomalously cold periods and large snowfalls in recent winters have been experienced in North America, Asia, and Europe [19], leading to increasing SCE for some areas [20]. Indications are that the quick warming of the Arctic is associated with changes in atmospheric circulation [21,22] and may be responsible for these anomalous events and areas of increasing SCE.

Monitoring SCE change in many parts of the world is difficult due to the lack of local observing networks, snow cover's spatial variability due to local conditions and other physiographic characteristics, frequent cloud cover, and confusion between lake ice and snow cover during the melt season [12]. In situ data provides detailed local data but remains extremely sparse across the globe. With satellite imaging, snow cover recognition is becoming more precise. There are now multiple satellite-based sensors, from thermal and microwave sensors to visible light sensors, that can capture snow cover. This research uses data from the Moderate Resolution Imaging Spectroradiometer (MODIS), which observes snow surface properties using solar illumination (visible and infrared wavelengths) in cloud-free periods and has been shown to be excellent for mapping SCE and duration [23,24]. The MODIS snow products have been validated by many studies [25–29], such as Hall and Riggs (2007), who found an overall absolute accuracy of the base 500 m resolution data to be about 93% [24]. This study uses MODIS data because it now has a greater than 23-year time span of consistent global-scale snow data, which have been used in numerous snow cover studies [8,10,16,20,25,30]. The MOD10C2 data set was used because it is a consistent data set, which minimizes cloud cover contamination, has over 23 years of reliable data, has been used in many other SCE studies [8,31–33] and has been validated, such as by Lei et al. (2011) who evaluated the snow identification accuracies of the MOD10C2 data to station data in northeast China and found the accuracy to be greater than 88%, with cloud cover being the main problem [34].

Different remote sensing studies have shown that SCE is broadly declining in different regions across the globe. Hammond et al. (2018) mapped global snow zones with MODIS data and found that between 2001 and 2016 for areas of snow cover, 5.8% declined in snow persistence while 1.0% increased in snow persistence. They also found that declining trends were greatest in the winter months [30]. Notarnicola (2020), using MODIS snow products, studied hot spots of snow cover change in mountain regions across the globe between 2000 and 2018 and found that 78% of observed areas were affected by snow decline and a snow cover area decrease of up to 13%, while above 4000 m only negative changes were observed [35]. In another publication, Notarnicola (2022) used a combination of snow cover data sets including MODIS and found that between 1982 and 2020 over global mountain areas an overall negative trend of $-3.6\% \pm 2.7\%$ for yearly SCE [36]. The season most affected by negative trends was winter. While most mountain ranges had negative trends in SCE, like the Alps, some mountain ranges, such as those in the northern high-mountain Asia region (Karakorum, Kunlun Shan, Pamir Mountains) had positive trends. Because the Northern Hemisphere has the highest percent (98%) of the world's snow cover between the Arctic and Antarctic circles [30], most snow cover studies have focused on the Northern Hemisphere. Kunkel et al. (2016) studied trends in extreme SCE in the Northern Hemisphere based on satellite observations for 1967 to 2015 and found an overall negative trend in SCE [37]. Using a multi-source remote sensing data set, Wang et al. (2018) found that between 2000 and 2015, the maximum, minimum, and annual average SCE in the Northern Hemisphere exhibited a fluctuating downward trend [38]. Using MODIS and AVHRR data, Hori et al. (2017) found an average decrease in SCE of 10 days/decade in the Northern Hemisphere since 1978 [39]. Using MODIS data, Eythorsson et al. (2019) estimated a decrease in Arctic snow cover frequency of 9.1 days/decade since 2001 [40]. Hernández-Henríquez et al. (2015) examined different latitudes and elevations of SCE declines in the Northern Hemisphere between 1971 and 2014 based on the NOAA snow chart climate data record and found the majority of statistically significant negative trends in the mid- to high latitudes [41]. Brown et al. (2021) analyzed snow cover trends for Canada (1955–2017) using the daily snow-depth-observing network of Environment and Climate Change Canada (ECCC) where results are broadly similar to previously published assessments showing long-term decreases in annual snow cover duration and snow depth over most of Canada [42]. Some large regions of snow cover have stayed relatively stable. Wang et al. (2017) used MODIS data and found no widespread decline in snow cover over on the Tibetan Plateau from 2000 to 2015 [43].

Many SCE studies have focused on snow onset dates and snow end dates [15] as well as the Northern Hemisphere's spring season [12,14,18,44] because with a high sun angle, spring snow in northern Canada, Alaska, and Siberia reflects extensive energy back to space that would otherwise potentially be absorbed and heat the planet further [2]. Shi et al. (2013), using the NOAA weekly snow cover maps, found that late spring/early summer SCE significantly decreased over the Arctic between 1972 and 2006 [45]. Derksen and Brown (2012), using the NOAA snow chart climate data records from April to June, found the Eurasia region set successive records for the lowest June SCE every year from 2008 to 2012 while North America set the June record 3 out of the 5 years (2008-2012) [46]. They also found the rate of loss of June snow cover extent between 1979 and 2011 ($-17.8\%$ decade$^{-1}$) is greater than the loss of September sea ice extent ($-10.6\%$ decade$^{-1}$) over the same period [8]. Brown and Robinson (2011), using the NOAA weekly SCE dataset, found that Northern Hemisphere spring SCE has undergone significant reductions over the past 90 years and that the rate of decrease has accelerated over the past 40 years. They also found that Eurasia had a significant earlier spring snow melt (March) than North America [14]. Musselman et al. (2021), using station data, found that in western North America (30 years+) snowmelt is increasing during the snow accumulation season [11].

This research differs from previous studies in that it analyzes changes in persistent SCE decline and increase globally and regionally. Although snow cover regions are quickly warming, changes in snow cover vary by region [14,18,21,47] and it is important to study the regional variation in a global context. There are few global scale SCE studies which focus on annual and seasonal areas increasing and decreasing the fastest. This study looks at areas of persistent decline and persistent increases in SCE globally and regionally from 2000 through 2022, and 2023 for the winter season. Most SCE change studies have generally focused on the spring season, when higher snow albedo feedbacks occur [14,46,48], but this study analyzes all four seasons and the snow-year (Northern Hemisphere: September to August of the following year, Southern Hemisphere: March to February of the following year).

This study uses the Mann—Kendall test to analyze persistent changes in SCE globally and regionally as well as applying the univariate differencing analysis to determine SCE change between the beginning and end of the study period. The Mann—Kendall test is a analytical tool commonly used to analyze changes in snow cover [8,11,35,40,41] while the univariate differencing analysis for change analysis is also used, but not as much as the Mann—Kendall test [8,49,50].

The objectives of the study are: (1) to analyze persistent trends in SCE throughout the four different seasons and annually (snow-year) at both the global and regional scales; and (2) to analyze changes in SCE through univariate differencing between the beginning (average of 2000 to 2004) and end (average of 2018 to 2022) of the period. The novelty of this research, in addition to studying the four seasons and snow-year, is that it uses the Z-values of the Mann—Kendall test to show the intensity of significant changes ($p < 0.05$, 0.01) as well as using the Mann—Kendall test significant values to filter the results of the univariate differencing analysis.

## 2. Materials and Methods

### 2.1. MOD10C2 Data Set

The MODIS/Terra Snow Cover 8-Day L3 Global 0.05 Degree Climate Modelling Grid (CMG), Version 61 (MOD10C2) data were used to map snow cover in this study. The MOD10C2 data have a MODIS sinusoidal projection, which is an equal-area projection and is appropriate for the global and regional analysis of SCE. The MOD10C2 data set has a spatial resolution of 0.05° and is an aggregation of MOD10A2 products with 500 m spatial resolution, which is an 8-day composite of MOD10A1 daily SCE maps [26]. The data were downloaded [6 January 2023 and 21 April 2023] from NASA's Earth Data web site (https://search.earthdata.nasa.gov/search). Version 61 data were downloaded and processed for the whole period from 24 February 2000 to 1 March 2023.

The 8-day composite is considered useful because persistent cloudiness limits the number of days available for surface observations in many regions, particularly at high latitudes [1]. However, MODIS snow products have some issues concerning cloud cover, the difficulty of detecting snow in forest areas, and topography that may impact the accuracy of the results [25,51,52]. Concerning extreme topographic changes, such as in mountainous terrain, several studies suggest that the use of MODIS snow cover products is particularly suitable to address the challenges in mountain areas [30,43,53]. The MOD10C2 product was chosen because the data set was developed for global studies and has been used in numerous studies [8,31,54–56]. The alternative MODIS global product is MOD10A1, which has a finer spatial resolution (500 m) and temporally resolution (daily), but this data set still has large data gaps and other issues [57] that require further processing.

MOD10C2 pixel values are the maximum percentage of snow cover (0% to 100%) for the pixel's area for eight continuous days [23]. The snow cover product is a Normalized Difference Snow Index (NDSI) created from MODIS band 4 (green) (0.545 μm to 0.565 μm) and band 6 (near-infrared) (1.628 μm to 1.652 μm). Cloud-cover contamination is the greatest deficiency of the MOD10C2 data set, while other issues, such as errors of commission, have been found to be very low [24].

Data were processed into seasonal and annual averages for December–January–February (12-01-02), March–April–May (03-04-05), June–July–August (06-07-08), September–October–November (09-10-11), and annually based on the snow-year or hydrologic-year (Northern Hemisphere: September to August of the following year, Southern Hemisphere: March to February of the following year) [20].

Because persistent cloud cover is the major issue with the MOD10C2 data, for this research snow analysis masks were created for each season and annually. An "analysis mask" is a means of identifying areas to be included in analysis. For each season (and annually), the maximum value for all 8-day snow cover files from 2000 to 2023 were calculated. The resulting maximum value snow cover file for each season (and annually) was then turned into a snow analysis mask where values of 100 = 1 and all other values = 0. The resulting snow analysis mask shows where there was at least one day of 100% snow cover in each season, and annually, between 2000 and 2023. These snow analysis masks were created so that for each season, only data within the masked area were analyzed; this way, clouds outside of the potential snow areas would not contaminate the data.

Clouds can cover areas of snow cover which can contaminate the data. To reduce cloud contamination, the MODIS cloud cover layer was added to the MODIS snow cover layer and when the 8-day cloud cover pixels plus 8-day snow cover pixels equaled a value of 100% and were completely surrounded by snow cover pixels with a value of 100%, it was assumed that clouds were covering 100% snow cover and for that 8-day period the pixels were classified as 100% snow cover. All other cloud pixels within the seasonal (annual) snow analysis masks were considered cloud contamination. A review of cloud contamination was performed on every 8-day snow–cloud combined file for the first and last five years of the data set and found the average of cloud contamination to be 2.9%. Because both Greenland and Antarctica are both primarily ice covered, these two regions were dropped out of the data set. Also, Greenland and Antarctica are in the high latitudes and thus a low sun angle during much of the year poses problems for global optical satellite data and analysis.

### 2.2. Mann—Kendall Test

This research used the Mann—Kendall test to determine if SCE values were increasing, decreasing, or staying the same for the global area of snow cover. The Mann—Kendall test is used to statistically assess if there is a monotonic upward or downward trend of SCE over time (2000–2023) [58–60]. A monotonic upward (downward) trend means that the variable consistently increases (decreases) through the period of the data set. The Mann—Kendall test analyzes the sign of the difference between later-measured data and earlier-measured data. There is no requirement that the measurements be normally distributed or that the trend, if present, be linear [61].

The Earth Trends Modeler in the Idrisi TerrSet 2020 software was used to analyze the MOD10C2 time series data with the Mann—Kendall test. For each pixel in the time series there are two resulting values: P and Z. The *p*-value provides the level of significance that there is a monotonic trend for that pixel and the Z-value is the number of standard deviations in the positive or negative direction of change. The Mann—Kendall test allows the determination of any apparent trend in a time series. A positive Z-value indicates an upward trend, a negative Z-value indicates a downward trend, and a zero Z-value indicates a lack of trend. Instead of using the Theil–Sen slope analysis as others have [20], the resulting Z-values were used to measure intensity of change.

The Mann—Kendall statistical test has been frequently used to quantify the significance of trends in meteorological time series [62–64] and snow cover analyses [20,48,54,65]. Using the Mann—Kendall test, this research performed global analyses of increasing and decreasing trends of SCE over time, which are considered significant at the 95th and 99th percentiles ($p < 0.05, 0.01$). At the regional level, trends were considered significant only at the 95th percentile ($p < 0.05$). To analyze change in SCE, two significant analysis masks ($p < 0.05, 0.01$) were created for each season and the annual data (significant values = 1, non-significant values = 0) based on the Mann—Kendall *p*-value for each season and annually. The resulting Z-values were multiplied by the significant analysis masks so that only significant pixels were evaluated for change (Z-value). When analyzing and mapping the results, only pixels that were found significant and had Z-values greater than 1 or less than $-1$ (more than one standard deviation of change) were used. To help illustrate the area of changing SCE, the area of France (second largest country in Europe) was used as an indicator of area. France's surface area of 549,087 km$^2$ was used and is based on data from the World Bank [66]. The actual areas of loss and gain were mapped by region.

### 2.3. Univariate Differencing

Univariate differencing is a technique to analyze one variable in a time series where a period at the beginning of the time series is subtracted from a period at the end of the time series. This method is used in a variety of satellite-based change detection studies [67], including MODIS-based studies [8,68,69]. The purpose of differencing using the beginning and end of the time series is to capture change that exists in the data. By subtracting the data between the first and last values, you eliminate a trend component, making the series stationary. To reduce anomalous years from the data, this study averaged the first five years of the SCE time series (2000–2004 (2001–2005 for the 12-01-02 season)) and the last five years (2018–2022 (2019–2023 for the 12-01-02 season)) and then subtracted the average of the first five years from the average of the last five years. This was performed for each season and for the annual data.

The univariate differencing technique does not determine significance of change, it just determines a static difference between the beginning and end of the time series. In addition to analyzing this static change, the results of the seasonal and annual univariate differencing were also multiplied by the 95th percentile ($p < 0.05$) Mann—Kendall test analysis mask (described above) to analyze significant change as well. The results of the univariate differencing indicate a percent change in SCE. For example, if a pixel was 100% covered at the beginning of the series but was 80% covered at the end, the univariate differencing result would be 20, indicating a 20% decline in SCE.

The univariate differencing method allows for the use of approximate change in days. For example, the 03-04-05 season has 92 days and if a pixel has an average of 100 (100% snow cover), then it had 92 days of snow cover ($1 \times 92 = 92$ days). If the pixel had an average value of 50%, then it would have 46 days of snow cover ($0.5 \times 92 = 46$ days). For univariate differencing analysis, if the pixel at the end of the period had a value of 90 and at the start it had a value of 100, then during the period SCE would have declined 9.2 days of SCE ($-0.1 \times 92 = -9.2$ days). These are approximate days because the 8-day maximum value composites create an uncertainty of how many actual days had snow cover. However, because of the averaging of the first and last five years and the use of

the same methodology throughout the time series, anomalies are not as prominent. The use of days of change creates a way to show the intensity of change and is not an exact measurement of days with and without snow cover. The number of days per season are as follows: 03-04-05 has 92 days, 06-07-08 has 92 days, 09-10-11 has 91 days, 12-01-02 has 90 days, and 1–12 has 365 days.

For the regional analysis, only results at the 95th percentile ($p < 0.05$) were evaluated and results were mapped in a latitude—longitude projection to show areas of change. The latitude—longitude projection does not have a constant scale and so this projection was used only for visual location of change. Statistics in all tables were made from a sinusoidal projection, which is an equal area projection, but a more difficult projection to see the location of change. The research framework can be seen in Figure 1.

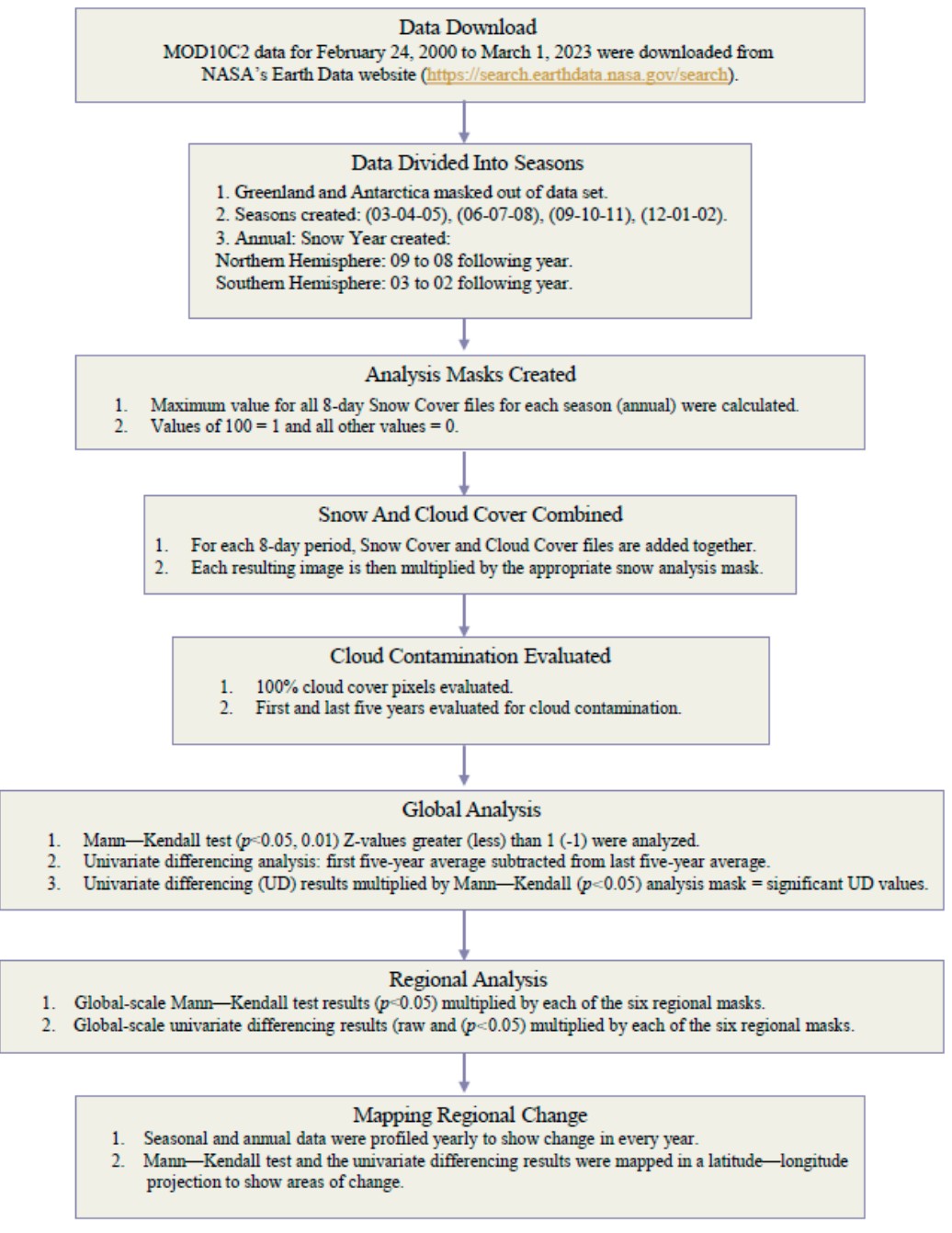

**Figure 1.** Methodological framework.

## 3. Results

### 3.1. Results Overview

Although both Antarctica and Greenland experience changes in snow cover, this research did not include these regions because they are primarily ice-covered and at high latitudes, which makes visible light imaging difficult much of the year. The primary focus of this research centered on changes in SCE on the non-permanent ice-covered regions. Even excluding Antarctica and Greenland, there still were a few ice-covered regions, such as mountain glaciers, included in the research. Excluding Antarctica and Greenland, the Northern Hemisphere had just over 97% of the world's SCE based on the annual snow analysis mask that captured the areas of the world, which had at least one day with 100% snow cover between 24 February 2000 and 1 January 2023. The regions with SCE that were analyzed, from largest SCE (based on annual total SCE) were: Asia (57.27%), North America (31.06%), Europe (9.04%), South America (2.02%), Australia/New Zealand (0.34%), and Africa (0.27%).

It is clear from the research that between the years 2000 and 2022, the world experienced broad changes in SCE. While some areas experienced a persistent increase in SCE, most of the change, as captured by the Mann—Kendall test and the univariate differencing analysis, was a persistent decrease in SCE. This research looked at both global changes and regional changes.

### 3.2. Global Analysis

The Mann—Kendall test was undertaken for all four seasons 12-01-02; 03-04-05; 06-07-08; 09-10-11, and at the annual level (snow-year, Northern Hemisphere: September through August of the following year, Southern Hemisphere: March through February of the following year). There are two results of the Mann—Kendall test: "P" probability of persistent change and "Z" intensity (measured in standard deviations from the norm). For each level of probability ($p < 0.05$ and $p < 0.01$) and a Z-value over 1 (under $-1$) during every season and at the annual level, the world experienced a greater persistent decline than increase in SCE, with declines averaging six times the area of increases (Table 1). Season 12-01-02 (Northern Hemisphere winters, Southern Hemisphere summers) experienced the greatest seasonal declines in both percent of change and area of change, with season 03-04-05 having the second largest areas of decline, and season 09-10-11 having the third largest decline (Table 1). At the annual level, areas of decrease minus areas of increase were as follows: at the 95th percentile ($p < 0.05$), the net area of loss was 5,275,456 km$^2$, which is larger than nine times the area of France and is 5.12% of the total global SCE, and at the 99th percentile ($p < 0.01$), the net area of loss was 1,432,324 km$^2$, which is more than two and a half times the area of France and is 1.39% of the total global SCE. Between 2000 and 2022, the world has persistently lost large areas of SCE.

Annually and for every season at each probability level, decreases in SCE were at least 70% of the total change, and above 80% most of the time, while increasing SCE was never above 30% of total change (Table 1). From the Mann—Kendall test analysis, globally areas of SCE between 2000 and 2022 were declining much faster than increasing. At the significance level of 95% ($p < 0.05$) annually between 2000 and 2022, the world experienced declining SCE 11 times more than increasing SCE with a net loss of over 5 million square kilometers. At the 99th percentile ($p < 0.01$), the world experienced declining SCE 15 times more than increasing SCE with a net loss of over 1.4 million square kilometers. (Table 1).

**Table 1.** Global Snow Cover Extent (SCE) change between 2000 and 2022 based on the Mann—Kendall Test [a].

| Season | Change | Statistical Significance | | | | | |
|---|---|---|---|---|---|---|---|
| | | $p < 0.05$ | | | $p < 0.01$ | | |
| | | Percent [b] | Area (km$^2$) | Frances [c] | Percent [b] | Area (km$^2$) | Frances [c] |
| 12-01-02 | Decrease | 86.66 | 3,138,440 | 5.74 | 82.05 | 848,284 | 1.55 |
| | Increase | 13.34 | 483,197 | 0.88 | 10.31 | 106,547 | 0.20 |
| | Difference | 73.32 | 2,655,243 | −4.85 | 77.68 | 741,737 | −1.36 |
| 03-04-05 | Decrease | 80.58 | 1,498,757 | 2.74 | 82.11 | 271,963 | 0.50 |
| | Increase | 19.42 | 361,150 | 0.66 | 17.89 | 59,241 | 0.11 |
| | Difference | 61.16 | 1,137,607 | −2.08 | 64.23 | 212,722 | −0.39 |
| 06-07-08 | Decrease | 82.15 | 837,589 | 1.53 | 79.51 | 197,656 | 0.36 |
| | Increase | 17.85 | 181,939 | 0.33 | 20.49 | 50,933 | 0.09 |
| | Difference | 64.31 | 655,650 | −1.20 | 59.02 | 146,723 | −0.27 |
| 09-10-11 | Decrease | 73.24 | 1,348,283 | 2.46 | 77.70 | 281,294 | 0.51 |
| | Increase | 26.76 | 492,528 | 0.90 | 22.30 | 80,755 | 0.15 |
| | Difference | 46.49 | 855,755 | −1.56 | 55.39 | 200,539 | −0.30 |
| Annual [d] | Decrease | 92.05 | 5,773,812 | 10.56 | 93.98 | 1,530,439 | 2.80 |
| | Increase | 7.95 | 498,356 | 0.91 | 6.02 | 98,115 | 0.18 |
| | Difference | 84.10 | 5,275,456 | −9.65 | 87.96 | 1,432,324 | −2.62 |

[a] Only values with a Z-score of >1 or <−1 were considered (at least one standard deviation from the norm). [b] Percent of total SCE change (area of persistent increase + area of persistent decrease = 100%). [c] To show the magnitude of the change, values are the number of Frances (area) that experienced SCE change. Area of France = approximately 549,000 km$^2$ according to the World Bank [66]. [d] Snow-year = Northern Hemisphere, September to August of the following year; Southern Hemisphere, March to February of the following year.

The univariate differencing results reflect the global changes seen from the Mann—Kendall test analysis. Based on both the raw values and the significant values ($p < 0.05$), SCE decreased broadly across the world, though a few areas experienced SCE increase (Table 2). Like the Mann—Kendall test, the season experiencing the greatest extent of decline was the 12-01-02 season, followed by the 03-04-05 season and the 09-10-11 season. Annually (snow-year), raw changes decreased three times more than increases and for significant changes, declines were 11 times more than increases. For every season, and annually, the significant changes saw a much greater ratio of decline to increase than the raw values. For significant values in both the 12-01-02 season and annually, declines of 20 or more days was the highest category of change, showing that these periods experienced intense levels of decline (Table 2).

**Table 2.** Univariate differencing [a] global seasonal SCE change between 2000 and 2022.

| | Raw Values [b] | | | Significant Values ($p < 0.05$) [c] | | |
|---|---|---|---|---|---|---|
| Change in Days [d] | Percent [e] | Area (km$^2$) [f] | Frances [g] | Percent | Area (km$^2$) | Frances |
| 12-01-02 | | | | | | |
| −20 | 14.28 | 2,424,386 | 4.43 | 45.15 | 1,382,073 | 2.53 |
| −10 | 29.77 | 5,052,411 | 9.24 | 31.44 | 962,750 | 1.76 |
| −5 | 29.25 | 4,965,115 | 9.08 | 12.98 | 397,420 | 0.73 |
| +5 | 13.70 | 2,325,961 | 4.25 | 3.01 | 92,070 | 0.17 |
| +10 | 10.76 | 1,826,024 | 3.34 | 4.74 | 145,204 | 0.27 |
| +20 | 2.24 | 380,246 | 0.70 | 2.67 | 81,778 | 0.15 |
| **Decrease** | **73.30** | **12,441,912** | **22.74** | **89.58** | **2,742,198** | **5.01** |
| **Increase** | **26.70** | **5,432,231** | **8.29** | **10.42** | **319,052** | **0.58** |

**Table 2.** *Cont.*

| | Raw Values [b] | | | Significant Values ($p < 0.05$) [c] | | |
|---|---|---|---|---|---|---|
| Change in Days [d] | Percent [e] | Area (km$^2$) [f] | Frances [g] | Percent | Area (km$^2$) | Frances |
| 03-04-05 | | | | | | |
| −20 | 0.54 | 74,431 | 0.14 | 2.37 | 34,193 | 0.06 |
| −10 | 13.80 | 1,886,412 | 3.45 | 32.59 | 470,704 | 0.89 |
| −5 | 58.13 | 7,948,927 | 14.53 | 48.88 | 706,025 | 1.29 |
| +5 | 21.75 | 2,974,264 | 5.44 | 10.75 | 155,310 | 0.28 |
| +10 | 5.52 | 755,191 | 1.38 | 4.13 | 59,706 | 0.11 |
| +20 | 0.25 | 34,317 | 0.06 | 1.27 | 18,352 | 0.03 |
| **Decrease** | **72.47** | **9,909,770** | **18.12** | **83.84** | **1,210,922** | **2.21** |
| **Increase** | **27.53** | **3,763,772** | **6.88** | **16.16** | **233,368** | **0.43** |
| 06-07-08 | | | | | | |
| −20 | 1.62 | 80,600 | 0.15 | 6.78 | 62,434 | 0.11 |
| −10 | 22.66 | 1,126,168 | 2.06 | 47.81 | 440,200 | 0.80 |
| −5 | 50.11 | 2,490,106 | 4.55 | 28.19 | 259,563 | 0.47 |
| +5 | 19.82 | 984,715 | 1.80 | 7.41 | 68,200 | 0.12 |
| +10 | 5.52 | 274,350 | 0.50 | 8.99 | 82,739 | 0.15 |
| +20 | 0.27 | 13,175 | 0.02 | 0.83 | 7595 | 0.01 |
| **Decrease** | **74.40** | **3,696,874** | **6.76** | **82.78** | **762,197** | **1.39** |
| **Increase** | **25.60** | **1,272,240** | **2.33** | **17.22** | **158,534** | **0.29** |
| 09-10-11 | | | | | | |
| −20 | 0.56 | 77,934 | 0.14 | 3.98 | 60,140 | 0.11 |
| −10 | 15.43 | 2,162,157 | 3.95 | 35.71 | 5,399,580 | 0.99 |
| −5 | 47.66 | 6,678,857 | 12.21 | 32.91 | 497,519 | 0.91 |
| +5 | 29.29 | 4,104,090 | 7.50 | 14.75 | 223,014 | 0.41 |
| +10 | 6.81 | 954,986 | 1.75 | 11.77 | 177,878 | 0.33 |
| +20 | 0.25 | 35,154 | 0.06 | 0.89 | 13,392 | 0.02 |
| **Decrease** | **63.65** | **8,918,948** | **16.30** | **72.60** | **1,097,617** | **2.01** |
| **Increase** | **36.35** | **5,094,230** | **9.31** | **27.40** | **414,284** | **0.76** |
| Annual [h] | | | | | | |
| −20 | 14.54 | 5,361,698 | 9.80 | 40.20 | 2,493,609 | 4.56 |
| −10 | 32.25 | 11,895,971 | 21.75 | 38.98 | 2,417,721 | 4.42 |
| −5 | 28.68 | 10,578,781 | 19.34 | 9.85 | 611,010 | 1.12 |
| +5 | 12.31 | 4,540,415 | 8.30 | 2.31 | 143,375 | 0.26 |
| +10 | 8.68 | 3,201,556 | 5.85 | 4.19 | 259,811 | 0.47 |
| +20 | 3.54 | 1,304,387 | 2.38 | 4.46 | 276,737 | 0.51 |
| **Decrease** | **75.47** | **27,836,450** | **50.89** | **89.04** | **5,522,340** | **10.10** |
| **Increase** | **24.53** | **9,046,358** | **16.54** | **10.96** | **679,923** | **1.24** |

[a] The average of the last five years of the SCE time series (2018–2022) minus the average of the first five years (2000–2004). For season 12-01-02: (2019–2023) minus (2001–2005). [b] These raw values are based on the results of the last five years subtracting the first five years, converted into days. [c] These values are the raw values multiplied by the Mann—Kendall 95th percentile ($p > 0.05$) mask. Thus, these values represent only the raw values that significantly ($p > 0.05$) persistently changed during the period. [d] Each season is made up of a certain number of days (03-04-05 has 92 days, 06-07-08 has 92 days, 09-10-11 has 91 days, 12-01-02 has 90 days, and 1–12 has 365 days). The resulting change in percent of the univariate differencing analysis is multiplied by a monthly day factor (92 days = 0.92), so a 10% decline in season 03-04-05 would = 10 × 0.92 = 9.2 days. [e] Percent of total area experiencing SCE change (area of increase + area of decrease = 100%). [f] The size (area) of decrease (increase). [g] To provide some context of the magnitude of the change, values here represent the number of Frances (area) that experienced decrease (−) or increase (+) in SCE between the first five years (2000-04) and the last five years (2018-2022) for the world. Area of France = approximately 549,000 km$^2$ according to the World Bank [66]. [h] Snow-year = Northern Hemisphere, 09-08 of the following year; Southern Hemisphere, 03-02 of the following year. Bold values are totals for each season and annually.

### 3.3. Regional Analysis

This research also evaluated the six major regions of the world with snow cover from most to least (based on percent of annual snow analysis mask): Asia (57.27%), North America (31.06%), Europe (9.04%), South America (2.02%), Australia/New Zealand (0.34%), and Africa (0.27%). While the global analysis evaluated SCE change at two significant values, 95th and 99th percentiles, the regional analysis only evaluated change at the 95th percentile.

Changes in SCE regionally varied in intensity and in temporal patterns. Regions that experienced the greatest area of loss and greatest area of increase at the annual level were in the same alignment with regional size as noted above (Asia to Africa), except for the annual increase where the Australia–New Zealand region had more than the South American region (Table 3). There was a net decline in SCE for every region in every season, except for Australia–New Zealand in the 06-07-08 season (Southern Hemisphere winter), which experienced a net increase of 10.02%. At the annual level, the Australia–New Zealand region also had an increase of 7099 km$^2$ or 3.61% of the total snow cover area of the Australia–New Zealand region. The 7099 km$^2$ increase is less than 0.1% of the snow cover area that had a net decrease globally (5,198,018 km$^2$) at the annual level at the 95th percentile ($p < 0.05$), thus globally this increase represents a small change.

**Table 3.** Regional SCE change 2000–2022 based on Mann—Kendall test Z-values [a]—($p < 0.05$).

| Season [b] Regions | % of Global | % Decrease | % Increase | % Decrease | % Increase | Difference | Area Change |
|---|---|---|---|---|---|---|---|
| | Total Seasonal SC [c] | Global SC [d] | Global SC [e] | Regional SC [f] | Regional SC [g] | Regional SC [h] | Regional SC (km$^2$) |
| 12-01-02 Africa | 0.32 | 0.34 | 0.09 | 7.40 | 0.31 | −7.10 | −10,081 |
| Asia | 58.45 | 66.46 | 51.07 | 8.09 | 0.96 | −7.13 | −1,836,823 |
| Australia-NZ | 0.03 | 0.10 | 0.03 | 27.07 | 0.93 | −23.13 | −3069 |
| Europe | 10.17 | 14.84 | 10.36 | 10.38 | 1.12 | −9.26 | −415,175 |
| N. America | 30.63 | 16.95 | 38.23 | 3.94 | 1.37 | −2.57 | −346,897 |
| S. America | 0.40 | 1.31 | 0.22 | 23.34 | 0.62 | −22.72 | −18,781 |
| Total | | | | | | | −2,630,826 |
| 03-04-05 Africa | 0.05 | 0.04 | 0.05 | 2.83 | 0.89 | −1.94 | −404 |
| Asia | 58.30 | 76.84 | 40.75 | 4.20 | 0.54 | −3.66 | −1,003,242 |
| Australia-NZ | 0.16 | 0.21 | 0.15 | 4.55 | 0.80 | −3.75 | −2605 |
| Europe | 7.98 | 8.38 | 30.41 | 3.44 | 2.92 | −0.52 | −19,547 |
| N. America | 32.65 | 12.83 | 28.29 | 1.25 | 0.67 | −0.58 | −88,880 |
| S. America | 0.86 | 1.70 | 0.34 | 6.67 | 0.32 | −6.35 | −24,331 |
| Total | | | | | | | −1,139,009 |
| 06-07-08 Africa | 0.14 | 0.40 | 0.16 | 18.23 | 1.53 | −16.70 | −3039 |
| Asia | 49.71 | 60.81 | 43.85 | 8.04 | 1.20 | −6.84 | −433,030 |
| Australia-NZ | 1.49 | 0.16 | 11.74 | 0.69 | 10.71 | +10.02 | +18,991 |
| Europe | 1.93 | 2.04 | 0.22 | 6.94 | 0.15 | −6.79 | −16,715 |
| N. America | 37.81 | 24.75 | 30.17 | 4.30 | 1.08 | −3.22 | −155,056 |
| S. America | 8.92 | 11.84 | 13.86 | 8.72 | 2.11 | −6.61 | −75,091 |
| Total | | | | | | | −663,940 |
| 09-10-11 Africa | 0.07 | 0.05 | 0.02 | 2.17 | 0.31 | −1.86 | −2575 |
| Asia | 59.36 | 57.41 | 61.23 | 3.23 | 1.26 | −1.97 | −472,278 |
| Australia-NZ | 0.16 | 0.65 | 0.03 | 13.15 | 0.19 | −12.97 | −8629 |
| Europe | 8.39 | 12.22 | 3.80 | 4.86 | 0.55 | −4.31 | −146,028 |
| N. America | 30.89 | 21.19 | 34.76 | 2.29 | 1.37 | −0.92 | −114,733 |
| S. America | 1.12 | 8.48 | 0.17 | 25.27 | 0.18 | −25.08 | −113,551 |
| Total | | | | | | | −858,258 |

**Table 3.** *Cont.*

| Season [b] Regions | % of Global | % Decrease | % Increase | % Decrease | % Increase | Difference | Area Change |
|---|---|---|---|---|---|---|---|
| | Total Seasonal SC [c] | Global SC [d] | Global SC [e] | Regional SC [f] | Regional SC [g] | Regional SC [h] | Regional SC (km²) |
| Annual [i] Africa | 0.27 | 0.28 | 0.12 | 10.10 | 0.39 | −9.71 | −15,531 |
| Asia | 57.27 | 64.24 | 48.30 | 10.95 | 0.74 | −10.21 | −3,417,688 |
| Australia-NZ | 0.34 | 0.19 | 3.55 | 5.64 | 9.26 | +3.61 | +7099 |
| Europe | 9.04 | 16.86 | 5.08 | 18.20 | 0.49 | −17.71 | −935,859 |
| N. America | 31.06 | 13.95 | 40.60 | 4.38 | 1.15 | −3.24 | −587,481 |
| S. America | 2.02 | 4.48 | 2.34 | 21.62 | 1.02 | −20.60 | −243,505 |
| Total | | | | | | | −5,198,018 |

[a] Values have a *p*-value of ($p < 0.05$) and Z-value of >1 (<−1), at least one standard deviation away from the norm. [b] Seasons: Northern Hemisphere winter (12-01-02), spring (03-04-05), summer (06-07-08), fall (09-10-11). [c] Percent based on the regional seasonal (annual) snow area divided by the global seasonal (annual) snow area. [d] Number of decreasing pixels in each region divided by the total number of decreasing pixels globally. [e] Same as d above, but for increasing values. [f] Percent based on pixels declining in the region divided by the total number of SCE pixels in that region during that season (annual). [g] Same as f above, but for increasing values. [h] Difference between decreasing and increasing regional SCE. Negative (positive) values = overall decline (increase) in percent regional SCE. [i] Annual = snow-year = Northern Hemisphere, September to August of the following year; Southern Hemisphere, March to February of the following year.

Regional areas with a small total SCE, such as Africa, Australia–New Zealand, and South America, experienced some of the largest regional changes, such as a net decline of 25.08% for South America in the 09-10-11 season or a 23.13% decline for the Australia–New Zealand region in the 12-01-02 season (Table 3). At the annual level, South America had the greatest regional decline in snow cover at 20.60%, followed by Europe (17.71%), Asia (10.21%), Africa (9.71%), and North America (3.24%). The only area experiencing a regional increase at the annual level was the Australia–New Zealand area at 3.61%. Four of the regions (mostly in the Northern Hemisphere: Africa, Asia, Europe, North America) had their greatest net area declines in the 12-01-02 season, while the other two (Southern Hemisphere regions: Australia–New Zealand, South America) had their greatest area declines in the 09-10-11 season (Table 3).

Concerning the temporal variations among the regions (Figure 2), based on $R^2$s, Asia had the most persistent decline ($R^2 = 0.8162$), followed by South America ($R^2 = 0.4704$), Europe ($R^2 = 0.2299$), Africa ($R^2 = 0.0829$), and North America ($R^2 = 0.0541$). Only the Australia–New Zealand area had a temporal increase, but only slightly ($R^2 = 0.0004$). Europe had the greatest inter-annual variation, followed by South America, while Asia had the least (Figure 2).

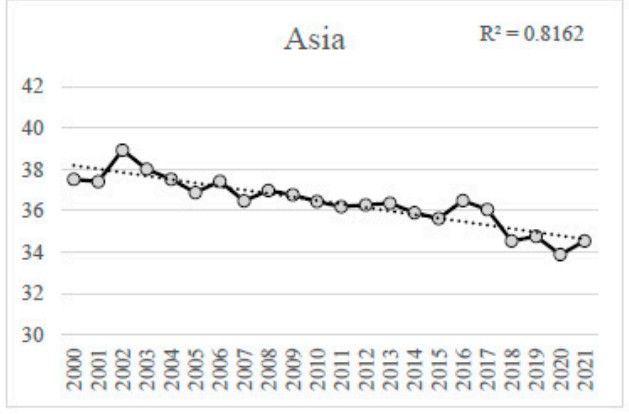
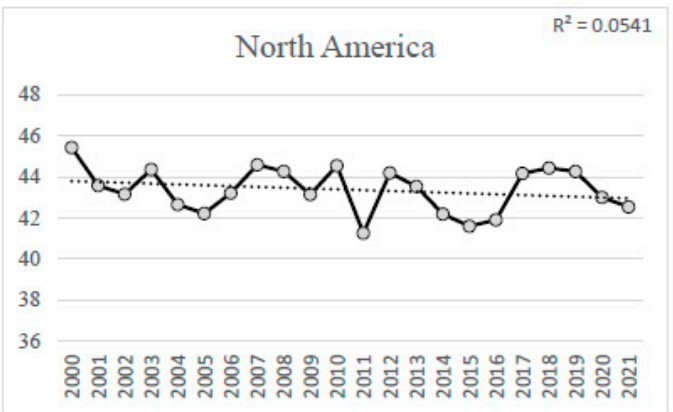

**Figure 2.** *Cont.*

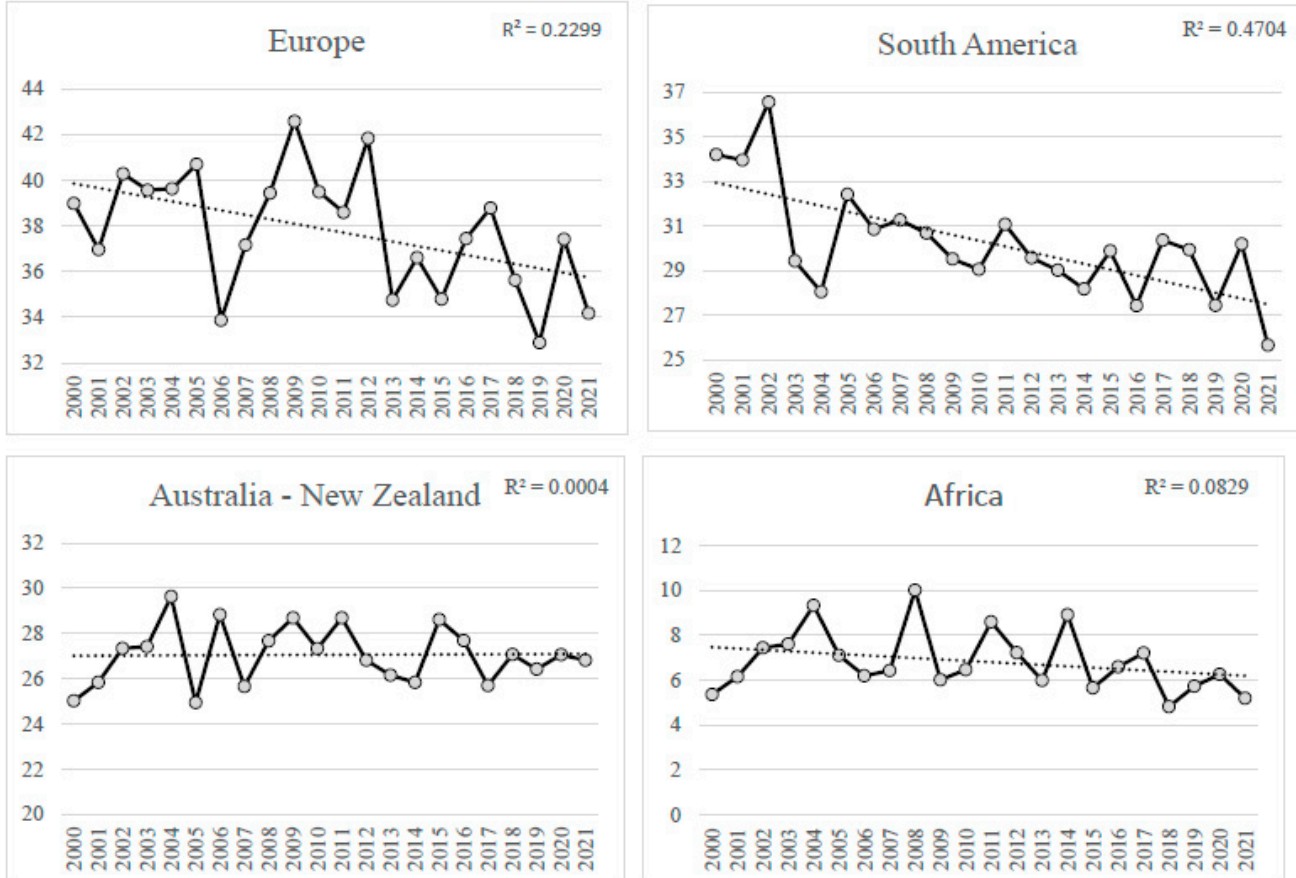

**Figure 2.** Annual snow-year SCE change based on Mann—Kendall test Z-values ($p<0.05$). Each point represents the snow-year average of all change pixels ($p < 0.05$) with a Z-value of 1 ($-1$) and greater (lesser). X-axis is the average annual percent of SCE in the region for all increasing and decreasing pixels with Mann—Kendal test ($p < 0.05$) and Z-value of 1 ($-1$) and greater (lesser). The X-axis for all regions has the same 12-value difference. The Y-axis denotes the snow years from 2000–2001 to 2021–2022.

### 3.3.1. Asia

Asia has the greatest extent of snow cover, more than any other region on Earth. In part because of its size, Asia is the region of the world experiencing a larger area of SCE loss that anywhere else in the world. Annually, and for all seasons, Asia had the largest areas of decreasing and increasing SCE and the greatest area of net change in the decreasing direction. For all seasons, Asia has over 49% of the world's total SCE, with no other region having more than 38% of global SCE (Table 3). Except for the 09-10-11 season, Asia had a higher percent of global SCE decline than it did of total SCE, meaning that Asia experienced a greater SCE decline than its size. For the 09-10-11 season, its percent of global SCE decline was only slightly less (57.41%) than its total of global SCE (59.36%). Concerning the percent of global SCE increase, except for the 09-10-11 season, Asia had a lower percent of total global increase than its total of the global SCE, indicating that it experienced less increases in SCE relative to its size of SCE. Again, the percent of global increased SCE for the 09-10-11 season was only slightly higher (61.23%) than its total global SCE (59.36%). Of the total SCE annually, Asia lost more than 14 times what it gained. Asia lost the largest net area of SCE in the 12-01-02 season, followed by the 03-04-05, 09-10-11, and 06-07-8 seasons (Table 3).

At the annual level (snow-year), Siberia experienced the largest area of loss, broadly across the North Siberian Lowlands and in southern Siberia from the West Siberian Plain to the Yablonnovy Range. After Siberia, at the annual level, the areas of the most intense loss were in northeast China (North China Plain and the Liaodong area) along with areas in

the Korean Peninsula and Japan. The third area of decline at the annual level occurred in Central Asia, stretching from the Tianshan mountains to the Caucasus, with broad areas of decline in Uzbekistan, western Kazakhstan, Iran, the Caucasus, and parts of the Russian Plain near Moscow. Although not as extensive, some areas in Asia experienced an increase in SCE, which occurred on the Chukotka Peninsula, throughout much of the Tibetan Plateau, and in the Himalayas into south central China, along with central Kazakhstan, and in the central Ural Mountains (Figure 3).

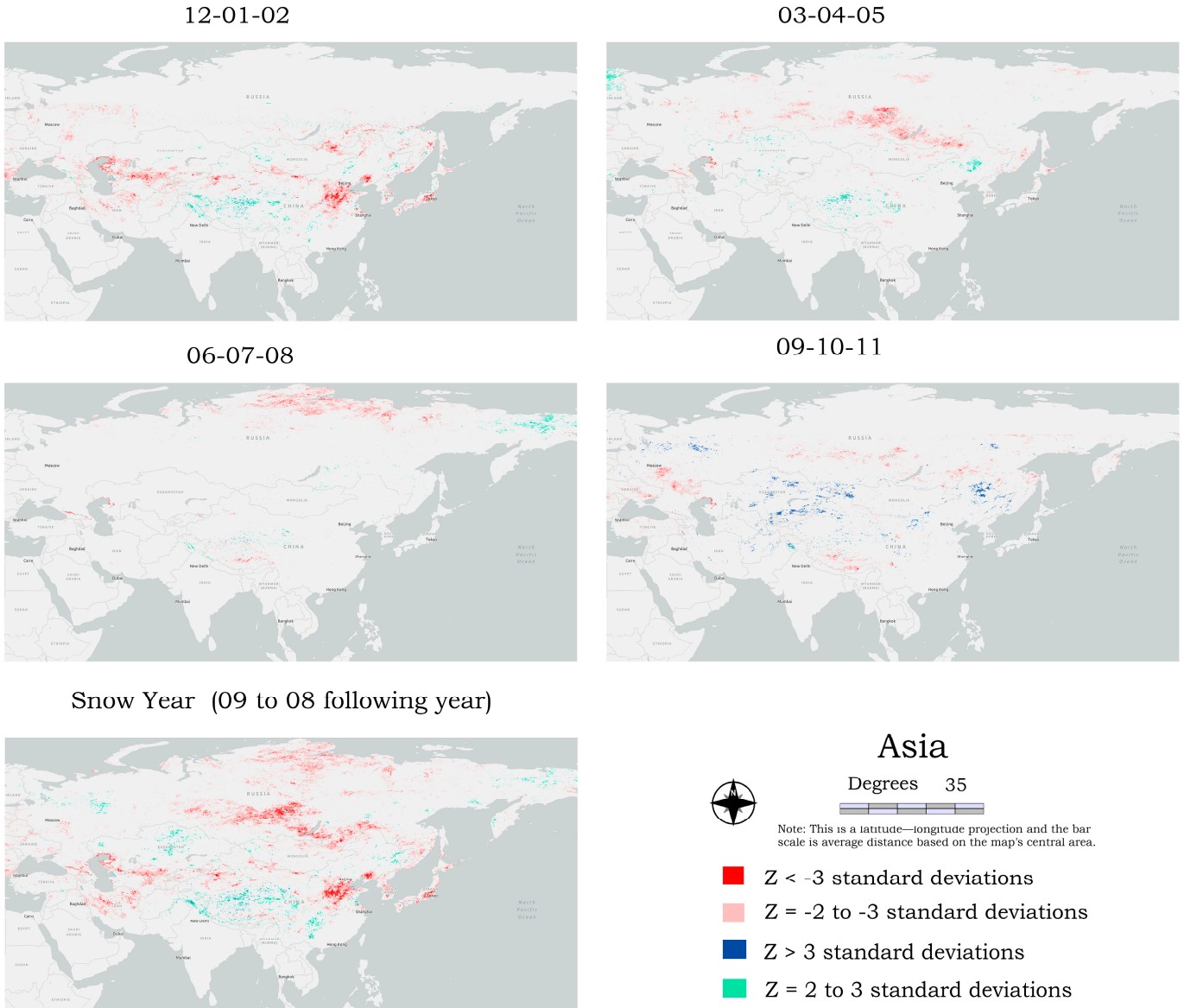

**Figure 3.** Snow cover extent change in Asia.

Changes in the 12-01-02 season are similar to the annual changes, except that Siberia does not experience much decline as it is winter, and the area is still cold and snowy. Most of the loss in Siberia occurs in the 03-04-05 and 06-07-08 seasons. Increases during the 12-01-02 season are similar to the annual change, with broad areas on the Tibetan Plateau increasing in SCE. In the 03-04-05 season, almost all the decrease is in the Siberian region, with some increases in northeast China and the Tibetan Plateau. The 09-10-11 season had the least amount of decrease, but some additional increases, especially in Northeast China and in Kazakhstan.

### 3.3.2. North America

The next most expansive region of SCE after Asia is North America. North America and Asia together account for 88.33% of SCE at the annual level and over 88% for every season (Table 3). Unlike Asia, North America's percent of global decreasing SCE is lower than its percent of global SCE for all seasons and at the annual level. In addition, its percent of increasing SCE is greater than its percent of global SCE for all seasons, except for 03-04-05 and 06-07-08. This means that North America is decreasing in SCE slower than the global average and it is experiencing increasing SCE faster than the global average, except for the seasons of 03-04-05 and 06-07-08. However, North America is experiencing more decreasing SCE than increasing SCE in every season and at the annual level, with the greatest area of loss occurring in the 12-01-02 season. At the annual level, the net loss of SCE between 2000 and 2022 is 587,481 km², which is about 85% of the size of the US state of Texas (695,660 km²).

Changes in SCE occurred mostly in the 12-01-02 and 06-07-08 seasons (regional winter and summer) and at the annual period (September to August of the following year). At the annual level, areas experiencing major declines in SCE are broadly across northern Canada, a scattering through Alaska, the central United States (especially in Kansas), and broadly in the US southwest and in the northeast (especially New York and New England). During the annual period, areas experiencing an increase in SCE includes a scattering along southern Canada, especially north of the Great Lakes and in the prairie provinces to the Rocky Mountains, along with northwestern and southeastern United States. During the 12-01-02 season, the greatest declines occurred in northeastern North America, especially southeast Canada, New York and New England, along with Kansas in the central United States. Most of the winter increases were in the western Canadian provinces of Alberta and British Columbia, the United States Pacific Northwest, Wyoming, and the US southeast. For its overall SCE, North America did not experience SCE change as intensively as other parts of the world (Figure 4).

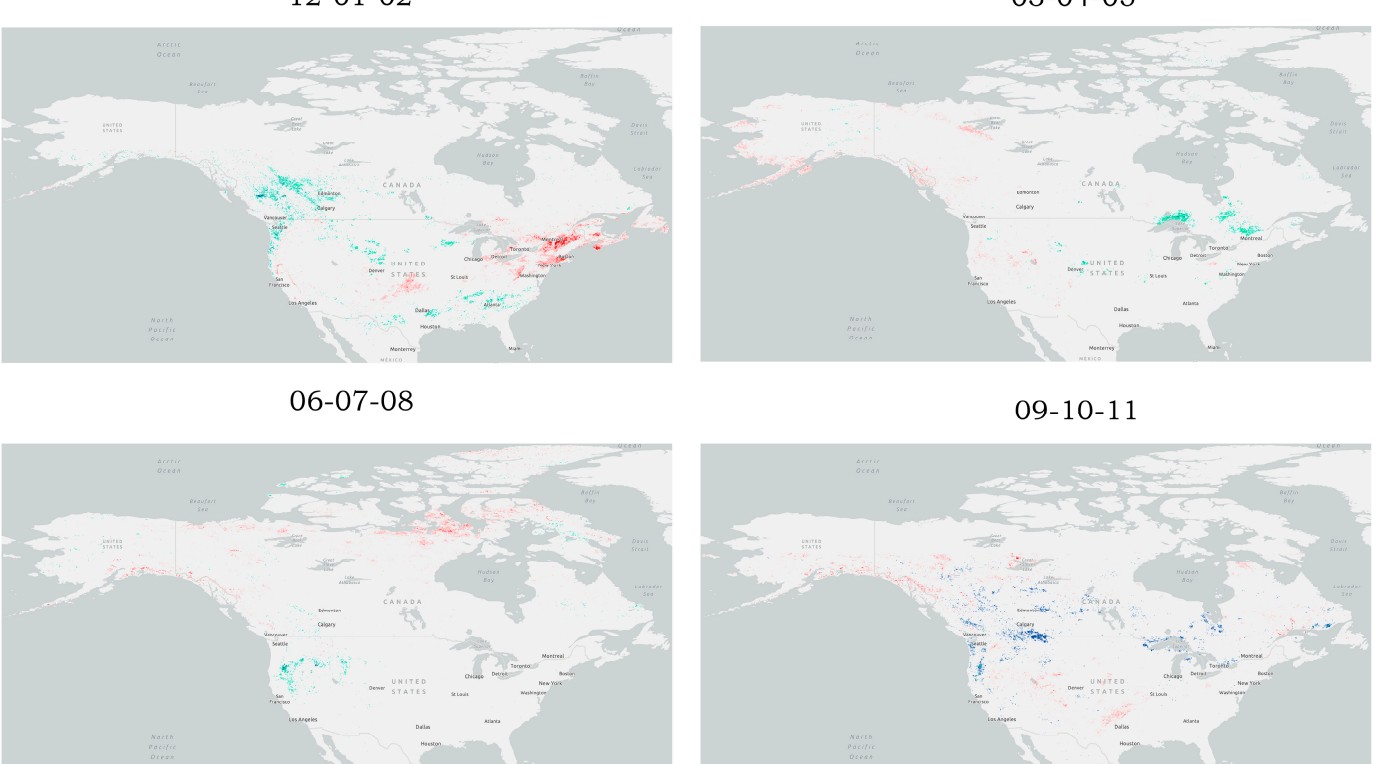

**Figure 4.** *Cont.*

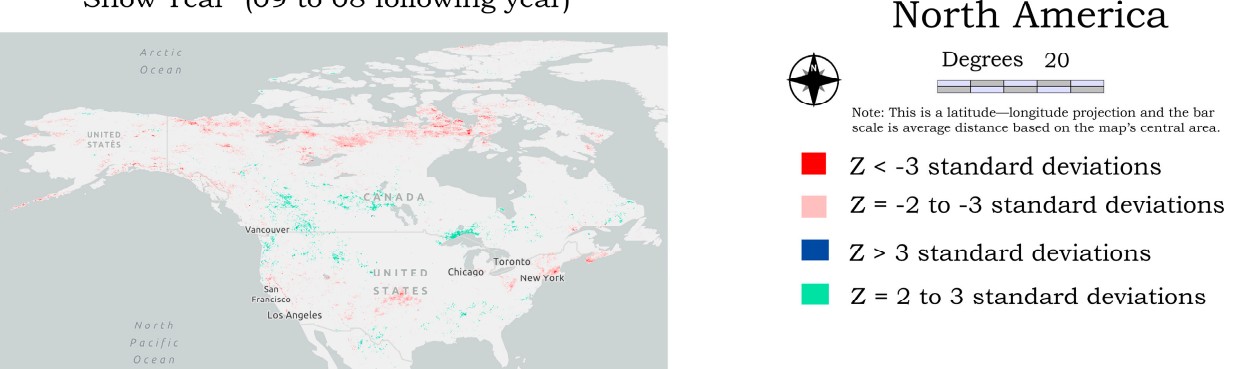

**Figure 4.** Snow cover extent change in North America.

3.3.3. Europe

The third-largest area of SCE was found in Europe, with 9.04% of global SCE at the annual level, over 10% during the 12-01-02 season, and less than 2% during the 06-07-08 season. Like Asia, Europe's percent of global decreasing SCE is higher than its percent of global SCE for all seasons and at the annual level, meaning that it experienced a disproportionate amount of SCE loss for its size. For two seasons (12-01-02, 03-04-05), but not at the annual level, Europe experienced more SCE increase than its share of SCE, but in all seasons and annually, Europe lost more SCE than it gained, a considerable loss relative to its size (Table 3). Temporally, at the annual level, Europe experienced a large inter-annual variation, greater than all other regions (Figure 2).

For this analysis, Europe did not include any portion of Russia, despite some definitions of Europe including western Russia to the Ural Mountains. Europe and South America experienced the greatest variation in SCE between the winter and summer seasons. Like North America, Europe experienced its greatest changes in the 12-01-02 and 06-07-08 seasons (regional winter and summer), though it did experience some areas of change in the 09-10-11 season. At the annual level, Europe lost more of its regional SCE than any other area besides South America, with a net loss of over 17% of its SCE between 2000 and 2022. At the annual level, Europe experienced a net loss of 935,859 km$^2$ of SCE, an area more than one and a half times the size of France (549,000 km$^2$).

Declines in SCE were much higher than increases in SCE for all seasons, especially in the 12-01-02 and 06-07-08 seasons as well as at the annual level. Declines in SCE occurred broadly throughout Europe at the annual level with a few areas of increase in northern regions of Europe, including Iceland, Ireland, Scotland, Norway, Sweden, and Finland. In the 12-01-02 season central and southeast Europe, from Hungary to Bulgaria, saw the greatest area of loss, while the only regions to experience an increase were in Ireland and Scotland. During the 03-04-05 season, there was a scattering of SCE loss in northern Italy, northern Poland, Latvia, and Lithuania, while there was a prominent increase in SCE in Norway and Sweden. In the 06-07-08, most of the decreasing SCE primarily occurred in the Alps (Figure 5).

3.3.4. South America

After Europe, the South American region was the fourth-largest area of SCE, with slightly over 2% of global SCE at the annual level, and just over 9% during the 06-07-08 season and under 0.5% in the 12-01-02 season. South America has the greatest amount of SCE in the Southern Hemisphere. At the annual level (Southern Hemisphere snow-year: March to February of the following year) South America had the greatest regional net loss of its SCE (20.60%) than any other region on Earth. It also had a greater local loss (percent) of SCE than anywhere else during the 03-04-05 and 09-10-11 seasons (Table 3). The region had an annual net loss of 243,505 km$^2$ of SCE from 2000 to 2022, slightly smaller than the size of Ecuador (276,841 km$^2$). South America had its greatest area net losses in the 09-10-11 season (Southern Hemisphere spring).

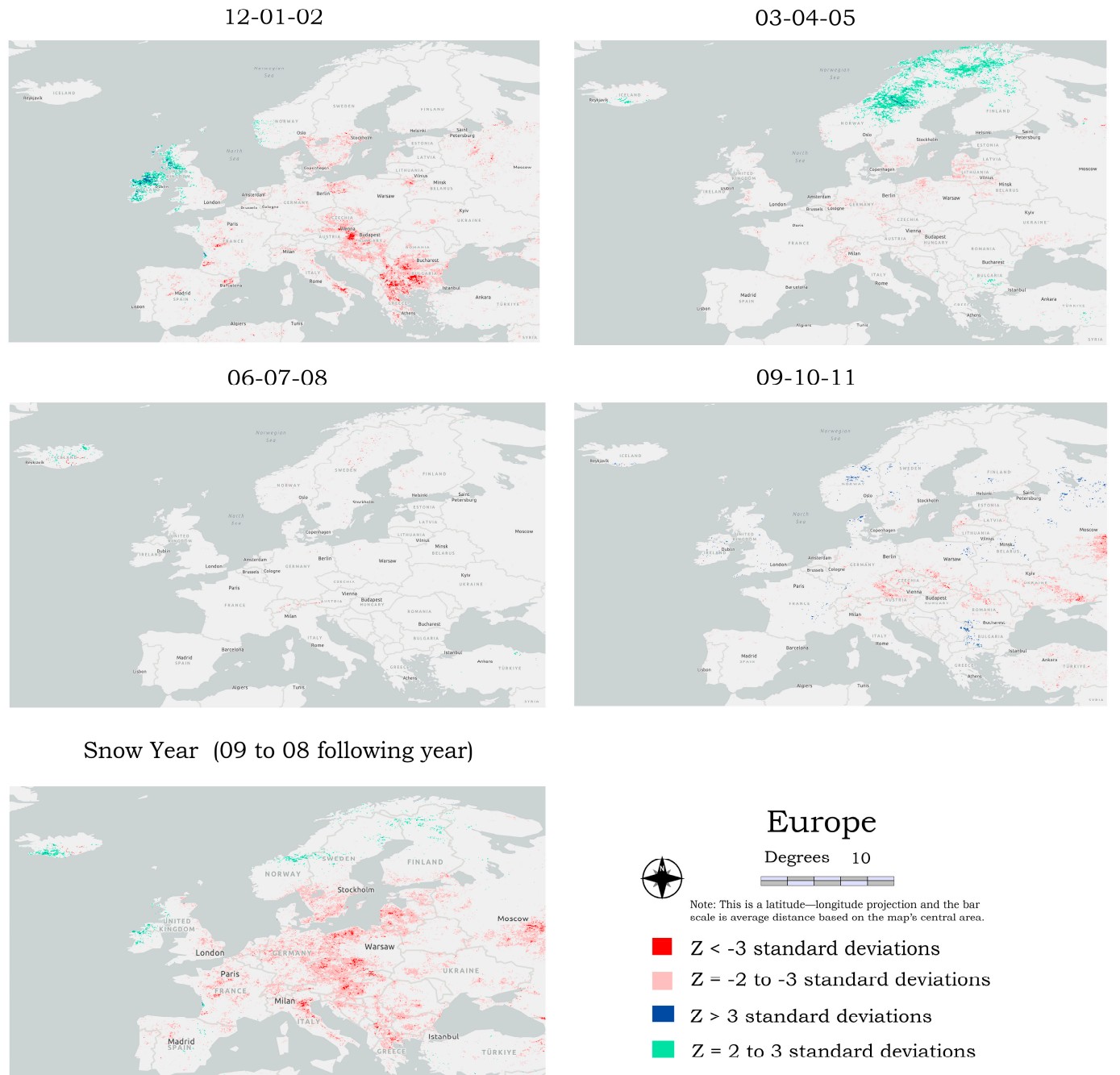

**Figure 5.** Snow cover extent change in Europe.

Most of the snow cover in South America is found in the southern Andes along the Chile–Argentina border, followed by Patagonia. At the annual level, there was SCE loss throughout the Andes and Patagonia, with the most intense areas of decline in the Andes between 29° and 37° S, and in Patagonia between 46° and 51° S and between 69° and 74° W. In the 12-01-02 (Southern Hemisphere summer) season, most of the SCE loss occurred in the Andes between 31° and 37° S. During the 09-10-11 (spring) season, the SCE loss occurred in the Andes between 31° and 37° S, and in Patagonia between 46° and 51° S and between 69° and 74° W, with almost no SCE increase. SCE increase in South America primarily occurred during the 06-07-08 (regional winter) season along the southwest coast of Chile (Figure 6).

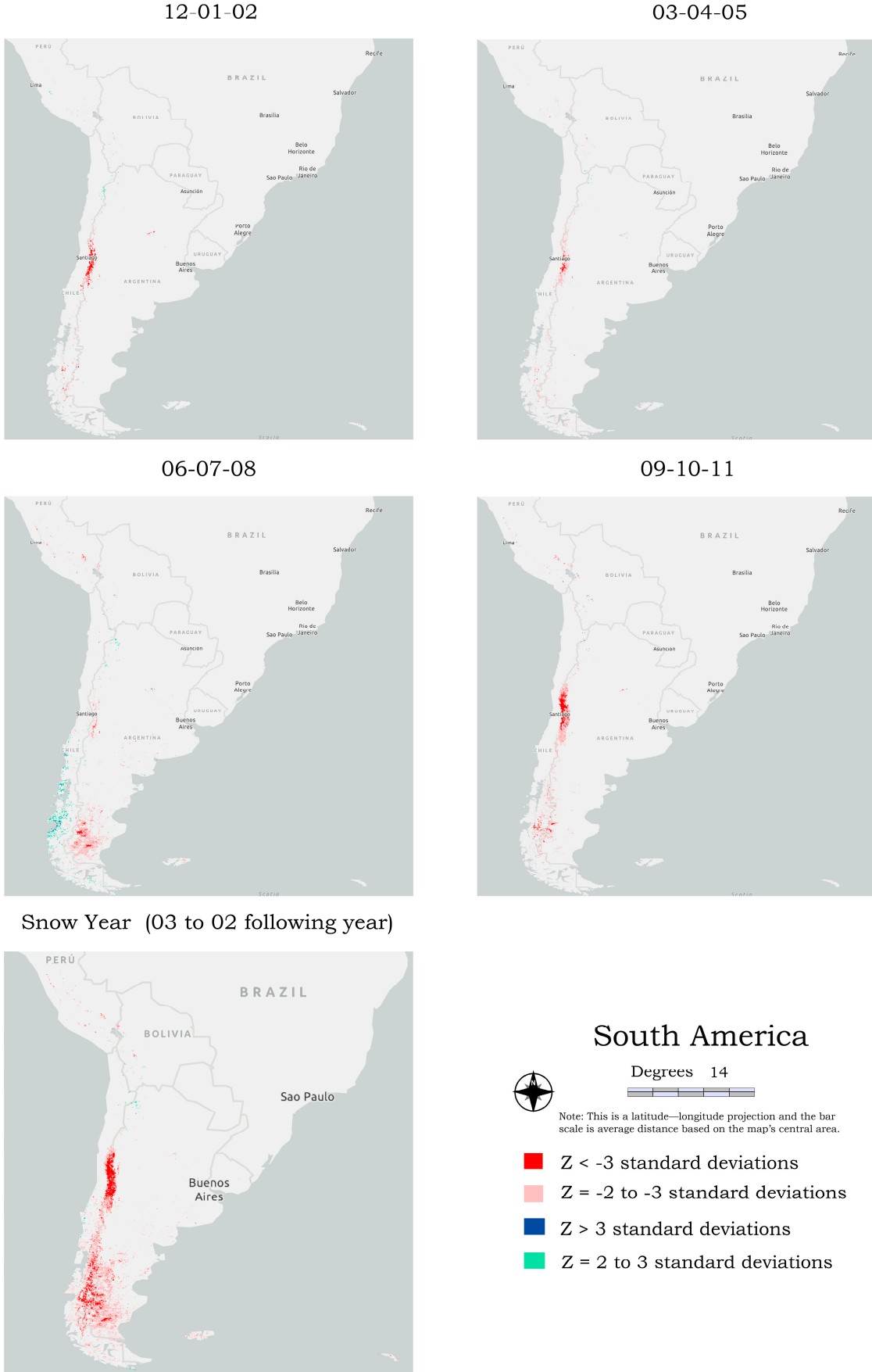

**Figure 6.** Snow cover extent change in South America.

### 3.3.5. Australia–New Zealand

The Australia–New Zealand region makes up a very small portion of the global SCE, having just 0.34% of the global SCE at the annual level and fluctuating from 0.03 to 1.49% from 12-01-02 to 06-07-08 (Southern Hemisphere summer to winter) seasons (Table 3). This is the only region which experienced some net increase in SCE. At the annual level, this region increased its SCE by 3.61% (7099 km$^2$), slightly smaller than the size of the Wellington region of New Zealand (8140 km$^2$). The only season with a net increase was the 06-07-08 (Southern Hemisphere winter) season. All other seasons experienced a net decline, though much smaller area than the 06-07-08 net increase. Increases occurred broadly throughout both the North and South islands of New Zealand, as well as broadly throughout Tasmania in Australia (Figure 7). The declines were primarily in the Southern Island of New Zealand in the mountains.

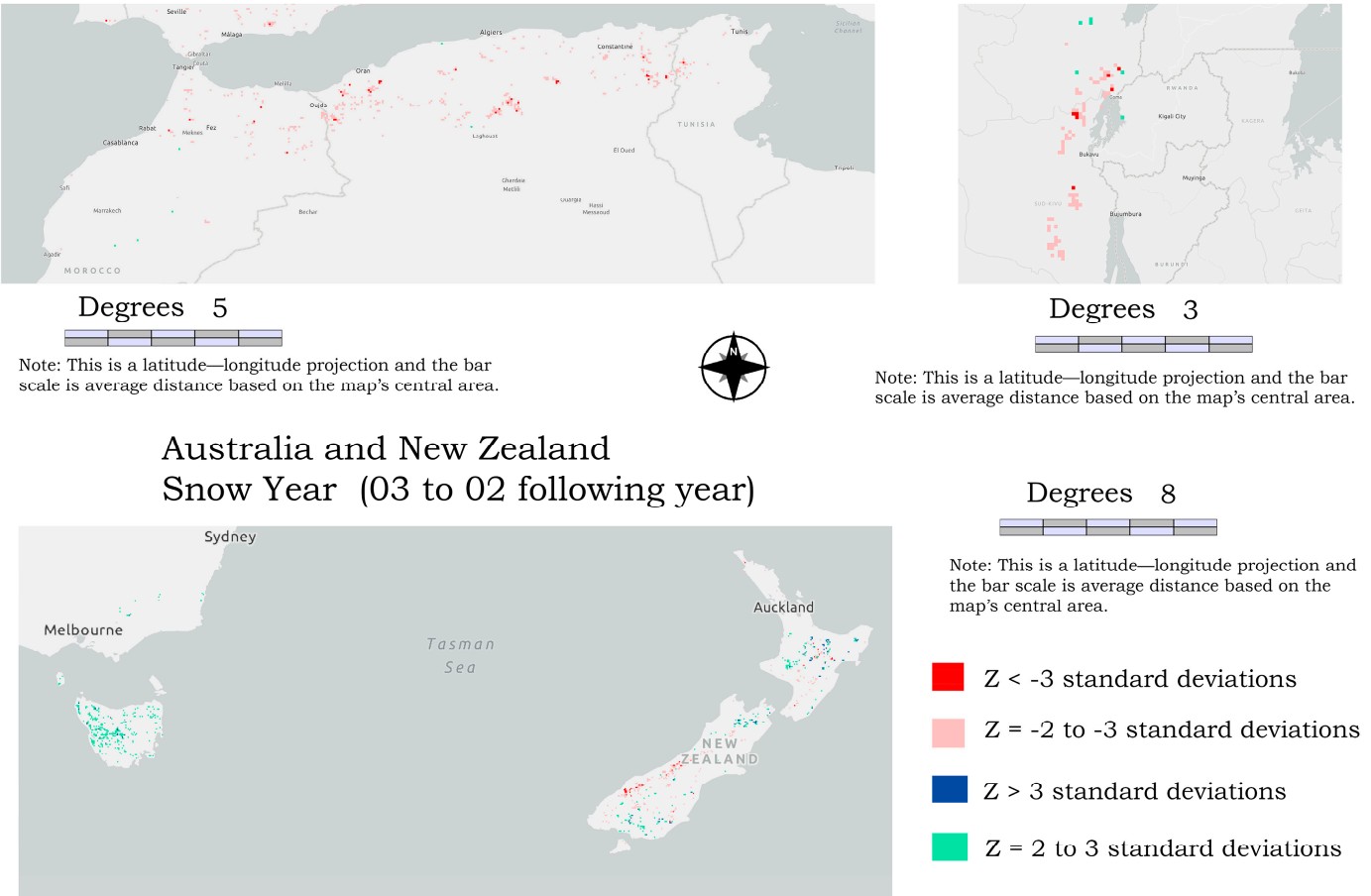

**Figure 7.** Annual snow cover extent changes in Africa and Australia–New Zealand.

### 3.3.6. Africa

The African region has the smallest percent of global SCE, at 0.27% at the annual level and ranging from 0.05 to 0.32% from 03-04-05 to 12-01-02. Snow cover in Africa is scattered in many places, often small patches isolated from other snow areas, such as in south and east Africa. The two main regions of snow cover are found in the Atlas Mountains in the north with over 80% of the continent's snow cover, based on the annual snow cover map, and in the Virunga mountains in central Africa. Both African regions experienced a decline in SCE. Together, both regions experienced a net decline in all seasons, and at the annual level it lost 9.71% of its local SCE (15,531 km$^2$) (Table 3, Figure 7).

Figure 8 shows the top 5% and 10% of snow cover decline based on the univariate differencing analysis at the 95th percentile ($p < 0.05$) (Table 3). For the Northern Hemisphere, the season of greatest decline was the 12-01-02 season and, for the Southern Hemisphere, the season of greatest decline was the 09-10-11 season. Table 3 shows that Asia and Europe have the largest areas of declining SCE and this figure shows these two regions also have the areas with the greatest intensity of decline. North America only has one major area of declining SCE at the top intensity, which is the southern New England area. South America has a large regional area of decline in the Andes, but the African and Australia–New Zealand regions did not have areas of intense SCE decline. Figure 8 shows the hot spots of SCE decline in the world between 2000 and 2022 (between 2001 and 2023 for the winter season).

### North America    12-01-02

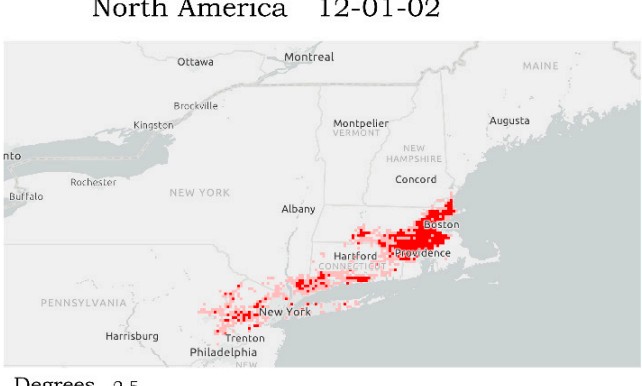

Degrees   2.5

### South America    09-10-11

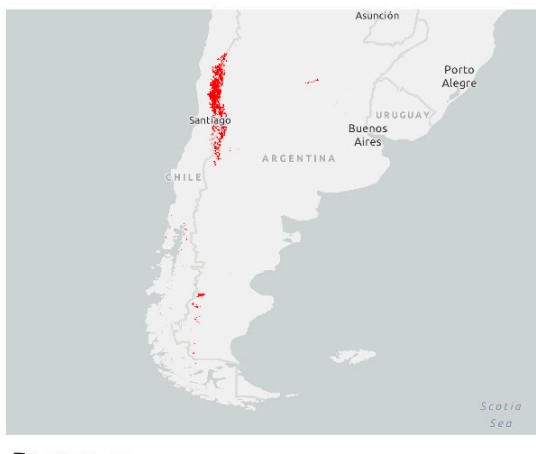

Degrees   11

Note: These maps are in a latitude-longitude projection and the bar scale is average distance based on the map's central area.

■ Top 5% of global snow cover decline

+ ■ Top 10% of global snow cover decline

### Europe 12-01-02

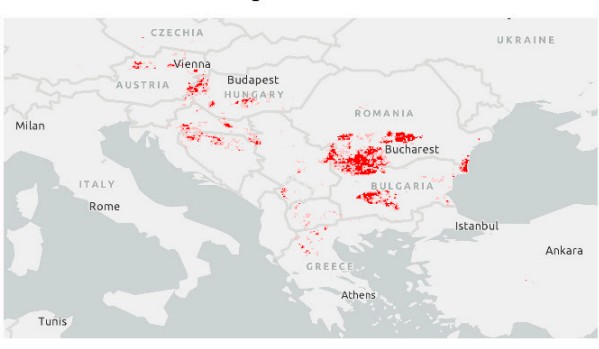

Degrees   5

### Asia 12-01-02

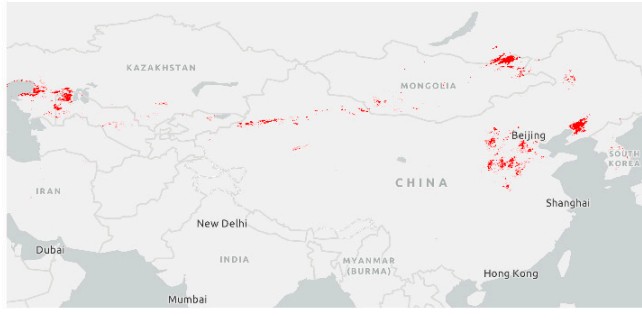

Degrees   15

## Hot Spots of Global Snow Cover Decline

Results based on the *p*<0.05 Univariate Differencing of the MOD10C2 data from 2000 to 2022.

**Figure 8.** Top 5% and 10% of SCE decline.

## 4. Discussion

Both the Mann—Kendall test and the univariate differencing analysis showed that the snow-covered portions of the world decreased during the first 23 years of the 21st century. Although some regions of the world have seen increases in SCE over the past 23 years, globally, declines were much greater than increases in all seasons and annually,

with declines being more than twice to more than ten times the area of increases for the different seasons. The Mann—Kendall test, with a significance level of 95% ($p < 0.05$) and a Z-value of greater than 1 (less than $-1$), showed that the snow-covered portion of the Earth had a net loss of 5,275,456 km$^2$, or 5.12% of its snow area size, or an area more than nine times the size of France, during the period from 2000 to 2022.

Although the time series used for this study is only 23 years long and is not at the length (30 years+) often used to analyze climate influences [70,71], Roessler and Dietz (2022) found that the 23 years of MODIS snow data (2000–2022) are already sufficient to determine significant trends for a considerable part of the observed areas globally [20]. The results of this study clearly show that changes in the 21st century are an extension of the declines in SCE seen in the 20th century [72]. There is documentation that the Northern Hemisphere has experienced SCE declines since the early 1920s [13].

Most of the world's snow cover exists in the Northern Hemisphere. Large-scale analyses of Northern Hemisphere SCE change have found a decrease in SCE of 1 million km$^2$ (or 4%) for the entire Northern Hemisphere compared to the long-term mean since 1966–2020 [73]. Another study found a global change in snow cover duration for the full hydrologic year (Northern Hemisphere: 1 September to 31 August of the following year; Southern Hemisphere: 1 March 2000 to 28 February of the following year) of $-0.44$ days/year between 2000 and 2022 [20]. Research into changes in SCE for global mountain ranges between 1982 and 2020 found an overall negative trend of $-3.6\% \pm 2.7\%$ for yearly SCE [36]. Using MODIS snow data from 2000 to 2018, it was found that around 78% of the global mountain areas are undergoing a SCE decline, with some areas declining up to 43 days and a SCE decrease of up to 13% [35]. Although changes in precipitation are noted as a cause of changing SCE, most studies point to warming surface temperatures driving the substantial reduction in the extent and duration of Northern Hemisphere snow cover [74]. These studies reflect the SCE changes found in this research.

This study's regional analysis shows similar comparisons with other SCE change analyses. As noted above, the Asian region had the greatest overall decline of SCE in the world. One of the major global areas in which a decrease in SCE occurred was Siberia, especially during the 03-04-05 (spring) and 06-07-08 (summer) seasons. Notarnicola (2020) noted a strong decline in snow cover mainly located in the southern and central parts of the region in the mountains [35], and Wu et al. (2023) has noted a decline in Siberian snow cover in the spring [75].

In the winter and annually, this research noted that central and eastern Japan has seen a decrease in SCE. Since the late 1980s, snow cover has been decreasing over the Japanese archipelago [76]. Broad areas in Iran saw a decline in SCE, and other research has noted that the Zagros Mountains experience a significant SCE decrease of $-20.7\%$ [35]. The Tianshan mountains of western China showed a decline in the central and eastern portions during the 12-01-02 season, and other research has shown that a warming trend in the region has led to declines in SCE in the central and eastern Tianshan mountains [77]. This research also showed an increase in the northern and western Tianshan mountains in the 09-10-11 season, and others have also noted this as an area of increasing SCE [75].

Concerning other areas of Asia increasing in SCE, Smith and Bookhagen (2020) analyzed a time series of high-resolution snow water equivalent data over the Tibetan Plateau from 1986 to 2016 and found positive trends in Karakorum, Hindu Kush, and Kunlun Shan during the winter and summer periods [78]. The research presented here also found winter and summer increases in SCE on the Tibetan Plateau. Multiple studies have shown that there appears to be no major decline in SCE on the Tibetan Plateau [43,79]. In the far northeast of Russia, in the Chukotka region, increases in SCE occurred in the 06-07-08 season, which other researchers have also noted [35]. Between 2000 and 2015, Liu et al. (2017) found an increasing trend in SCE in central Kazakhstan in areas similar to the increasing trends found in this research in the 09-10-11 season and annually [80].

The second-largest area of SCE is North America. Unlike Asia, North America did not experience a decline in SCE as intensively as Asia or any of the other regions, except for the Australia–New Zealand region, which saw some net increases. Annually and for every season, except the 03-04-05 season, North America experienced the least local change of any region, and for the 03-04-05 season, only one other region experienced less change (Table 3).

At the annual period regions in northern and northwestern Canada, along with multiple regions in Alaska, experienced declines. Research in Alaska has presented similar trends, with a strong decrease in SCE [81]. When averaged across the state, the disappearance of snow in the spring has occurred from 4 to 6 days earlier per decade, and snow return in fall has occurred approximately 2 days later per decade [81]. This change appears to be driven by climate warming rather than a decrease in winter precipitation, with average winter temperatures also increasing by about 2.5 °F [81]. Northern Alaska has been experiencing a decline in SCE over the first 17 years of the 21st century [82]. Brown et al. (2021) analyzed snow cover in Canada from 185 stations and found an SCD decrease of −1.68 days/decade in the boreal autumn and an increase of +0.28 days/decade in the boreal spring [42]. This research also noted increases in Canada during the 03-04-05 season. Increased snow north of Lake Superior might be due to lake-effect snowfall, which is increasing around the regions of Lake Michigan and Lake Superior [83]. One of the few contiguous areas of concentrated declines is in northeastern North America. Decreasing snow in the Northeast between 2000 and 2017 has been reported by others [8]. Across New England, there once were numerous local downhill skiing and sledding areas, but without reliable snow cover, many have closed, especially in southern New England [84,85], an area of snow cover that this research shows is quickly disappearing.

The third major area of snow cover is Europe, which for every season experienced a greater percent decline of SCE than its percent of global SCE and a greater increase for two seasons (12-01-02, 03-04-05) (Table 3). Europe also had the greatest interannual fluctuation of change (Figure 2), where the predominant winter teleconnection pattern over the Euro-Atlantic region, i.e., North Atlantic Oscillation (NAO) is a driving factor for the interannual variability of European snow cover [86], and the inter-annual variability means some years can deviate tremendously from average snow cover patterns in Great Britain [87].

This research shows that much of continental Europe, especially the central and eastern areas, have lost SCE, particularly during the winter season. Tomczyk, (2021) has noted that lowland areas of central and northern Europe have experienced a reduction in snow cover duration, and central Europe has experienced several mild winters since the 1990s, with the winter of 2019/2020 being extremely warm and snowless [88]. The duration of the snow season in Europe has decreased by up to 25 days in western, northern, and eastern Europe due to earlier spring melt [89]. Significant negative trends are found in the Alps and the Carpathians [36]. For the European Alps, a recent study found that over all stations and all months, 87% of the trends were negative and 13% positive [90]. Concerning areas of increase, there is a prominent increase in SCE in Norway and Sweden during 03-04-05 season, which other research has documented [20]. During the winter season, much of Ireland and western Scotland also indicate an increase in SCE. Occasionally, cold air from Siberia brings unusual winter snow to the British Isles [91].

Excluding Antarctica, most of the SCE in the Southern Hemisphere is restricted to high altitude areas in the Andean region [92], though in the southern region of Patagonia there is also some SCE. The Andes stretch north and south along western South America, and its snowpack is the primary source of water for many communities. Like Europe, the change in SCE in South America experienced some interannual variability, though it was not as extensive as in Europe (Figure 2). The interannual variation in South America is caused by El Niño Southern Oscillation (ENSO) events [92]. Using Landsat images across a north–south transect of approximately 2500 km (18–40° S) between 1986 and 2018, it was found that SCE declined across the entire study area at an average rate of about

−12% per decade [92]. This research discovered intense areas of decline (2000–2022) in the Andes and Patagonia and, at the annual snow-year level, SCE declined by 20.6%; a similar decadal rate of decline was found between 1986 and 2018 [92]. The greatest seasonal decline occurred in the 09-10-11 (spring) season, with a net decline of 25.08% (Table 3, Figure 6). According to MODIS estimates over the period 2000–2016, the annual SCE shrunk in the Andes by about 13% around latitude 34° S [93]. Decreasing snow trends may be partially attributed to increases in surface air temperature [94]; however, precipitation seems to be a main driver [95]. Precipitation changes are being driven by ENSO events [93]. The poleward migration of the westerly winds has led to precipitation drops at Andean mid-latitudes, leading likely to decreasing SCE [96]. Declines in SCE in Patagonia have also been documented [96,97], including a significant decreasing trend of SCE of 19% for Patagonia's Brunswick Peninsula for a 45-year period (1972–2016), which is attributed to a significant long-term warming of 0.71 °C at Punta Arenas during the extended winter (April–September) [98].

The last region of declining SCE was in Africa, which has the lowest percent of global SCE at 0.27% annually and varies between seasons from 0.05% (03-04-05) to 0.32% (12-01-02). As noted above, snow cover in Africa is scattered in many places, with only two areas of relatively continuous snow cover, North Africa and Central Africa. Both regions were found to have a declining SCE (Table 3), though areas of decline were spread out and not strongly spatially clustered (Figure 7). Past research in the Atlas Mountains found no evidence of a significant long-term trend; however, it was discovered that snow cover increased in February–March and decreased in April–May for their research period of 2000–2013 period [99]. There is a dearth of published research into SCE change in snow areas of Africa. Much of the published research is on famous individual mountains like Mt. Kenya and Mt. Kilimanjaro, such as the snow cover area of Mt. Kilimanjaro largely decreased from 10.1 km$^2$ to 2.3 km$^2$ between 1984 and 2011, which corresponds to a 77.2% reduction [100].

The only region to exhibit more increasing SCE than decreasing for a season, or annually, was the Australia–New Zealand region. The Australia–New Zealand region makes up a very small portion of the global SCE, having just 0.34% of the global SCE at the annual level (Table 3). At the annual level, this region increased its SCE by 3.61% (7099 km$^2$) and spatially these increases are seen across Tasmania along with the North and South Islands of New Zealand, while prominent declines are found in the mountainous South Island, especially in the central to southwestern regions. Other researchers have found that Australia and New Zealand show an overall increasing SCE, though there were no large areas with significant trends [35]. New Zealand has considerable interannual variability [101] and the increases found here may be due to fluctuations in SCE. According to Planet Ski news [102], both Australia and New Zealand had heavy winter snows in 2022 [102].

A 16-year (2000–2016) time series of daily snow-covered area, derived from MODIS imagery, was used to analyze SCE change for New Zealand's largest catchment, the Clutha Catchment. In contrast to other regions globally, no significant decrease in snow cover was observed, but substantial spatial and temporal variability was present [101]. In Tasmania, the incidence of snow fluctuation between 1983 and 2013 at Mt Field was shown to have no overall trend [103].

## 5. Conclusions

This study used global scale snow cover data (MOD10C2) from February 2000 to March of 2023 to analyze how snow cover has changed over the first 23 years of the 21st century. Two methods were used to analyze the data, univariate differencing and the more commonly used Mann—Kendall test. Novel to this research is the use of the Mann—Kendall Z-value (standard deviations of change from the norm) to determine intensity of snow cover change and the use of the Mann—Kendall *p*-value ($p < 0.05$) to filter the results of the univariate differencing.

From this research and that of others, it is clear that the world is quickly losing its snow cover. The first 23 years of the 21st century show that changes in snow cover continue along a path of decline, which has been happening for the past 100 years. During the past 23 years, snow cover has been broadly disappearing around the world, especially in Asia, Europe, and South America, along with areas in North America and Africa. Snow cover in Asia is declining faster than the global average, and snow cover loss is extensive throughout much of the region. Europe and South America are also losing snow cover faster than the global average. North America, the world's second-largest area of snow cover, is experiencing a slower decline with multiple areas of increase. Despite its slower pace, snow cover is also declining in North America and quickly in the New England region. Snow cover in Africa is not as concentrated as in other parts of the world, but the snow cover here is also declining. The only region with more increasing snow cover than decreasing was the Australia–New Zealand region, but the increases were slight and with the regional variation the future direction of change can go in either direction.

With global greenhouse gas emissions continuing at a record pace [104], the trend of decreasing SCE will continue. The next step in this research will be to explore the relationship between land surface temperature and changes in SCE, using the MOD11C3 land surface temperature data with the MOD10C2 snow cover data used in this paper, and potentially also exploring the use of the MOD10A1 data.

**Funding:** This research received no external funding.

**Data Availability Statement:** Data available upon request.

**Acknowledgments:** Support from the Office of Research and Creative Activities and the Department of Geography and Sustainability at Salem State University. Additional help from Christopher Yakes from Salem State University's Math Department.

**Conflicts of Interest:** The author declares no conflict of interest.

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
