# Peer review of "Global and Regional Snow Cover Decline: 2000–2022"

_climate, doi:10.3390/cli11080162_

Round 1
Reviewer 1 Report
See the attachment for details.

Minor editing of English language required.
Author Response
Comment 1: Paragraphs 1 and 2 can be merged and simplified. The description of glaciers can be deleted since this paper mainly concentrated on snow cover.
As suggested, I have combined the two paragraphs and have deleted the discussion of ice. Condensing the discussion and eliminating 10 lines.
Comment 2: In Paragraph 3, the remote sensing sensors used in SCE research and the reasons for choosing MODIS need to be complemented. In addition, the author stated “There are few global scale SCE studies”, what are the results of SCE studies on other scales, such as hemispherical scales? This part needs to be reflected in the introduction.
As suggested, I added information as to why the MODIS data was chosen and I have increased the introductory discussion about other findings concerning snow cover change. This was a clear weakness of the first submission.
Comment 3: There is no foreshadowing for the introduction of the M-K test and improvements are needed. Lines 85−90 are meaningless and can be deleted. The results of other studies using the M-K test and Univariate Differencing to explore changes in SCE in recent decades can be added here.
As suggested, I have deleted lines 85-90 and I have added more discussion about the Mann-Kendal and Univariate Differencing methods in the introduction and in methods section.
Comment 4: Lines 111−119, why did you use 8-day composite instead of daily snow products? Daily products have a higher temporal resolution, but data gaps may be relatively high. This should be explained in the manuscript.
As suggested, I have added some discussion about the data set chosen.
Comment 5: Line 149, why calendar year can “averaging reduces the impact of anomalies created by cloud cover”? I don't think this is an advantage compared to the hydrological year. Please support your statements with appropriate references
This is a great comment. I agree that the hydrologic year is better, and I have re-processed the data and have re-run the annual analysis based on the snow year: Northern Hemisphere: September to August of the following year and Southern Hemisphere: March to February of the following year. I decided to start the snow year in September because this is when snow begins to fall in Siberia and high latitudes. Some hydrologic years begin in October.
Comment 6: Section “2.4. Regional Analysis” is unnecessary and can be deleted.
As suggested, I have deleted section 2.4.
Comment 7: The conclusion needs to be re-summarized. Lines 689−696 can be deleted. Don't use the word "I" in scientific writing. The conclusion should indicate the purpose, method, and conclusion of the paper, as well as shortcomings and prospects.
I have rewritten the Conclusion based on the suggestions.
Other minor comments:
- The number of keywords is too much, please reduce it to less than 5.
done
- Lines 92−94, you don't need to explain “different seasons” and “annually” since it's common sense.
Reduced this discussion
- Line 95 “2000-04”, you mean 2000 to 2004? Please use the full expression.
Done
- Lines 120−131 can be deleted since it’s a common-sense acknowledgment. Lines 133−144 can be simplified, and can be placed in the introduction.
Has been simplified
- Figure 1 needs to be redrawn. Please delete the text and leave the keywords.
Have redone Figure 1.
Export images, not screenshots. Refer to flowcharts from other literature.
Images were exported in high resolution tiff format and zipped with the text. Screenshots were just done to put images into text.
- Use the “Times New Roman” font for the text in Figure 2. 7. Figures 3−8 require the addition of a scale and compass.
Done
- Lines 679−680, “The storm… in 20 years” can be deleted.
Deleted
Thank you for the helpful comment, they have greatly improved the manuscript.
Reviewer 2 Report
1. Methodological Framework is cumbersome, many descriptions in the methodological framework overlaps with the main text, which is unnecessary. A detailed description should appear in the main text.
2. The quality of all figures and tables is poor. And the notes below the table should be concise.
3. ‘Mask’ is generally considered to be removed, for example, ‘water mask’ means to remove water bodies, so ' snow mask ' means to remove snow, but what you should want to express is snow cover, so ' snow masks ' is not suitable.
4. Lines 160-166, the expression is not clear. Lines 160-162 say ‘When areas with potentially 100% snow cover are obscured with partial cloud cover (less than 100%), the cloud cover layer is added to the snow cover layer and the pixel value result equals 100%’. How to understand ‘potentially 100% snow cover’,’ partial cloud cover (less than 100%)’ and ‘pixel value result equals 100%’.
5. The description of MOD10C2 data should be based on the ‘MODIS/Terra Snow Cover 8-Day L3 Global 0.05Deg CMG, Version 61’ user guide, and this reference may be more suitable than ‘MODIS snow products user guide for Collection [38]’.
6. In Section 2 Materials and Methods, title 2.4 Regional Analysis is not suitable, because it is not a method.
7. According to the description in the text, this study should evaluate six major regions of the world with snow cover (Asia, North America, Europe, South America, Australia-New Zealand, and Africa) rather than five.
8. References 18, 22, 94 need to pay attention to formatting.
Further specialization is needed in English expression
Author Response
- Methodological Framework is cumbersome, many descriptions in the methodological framework overlaps with the main text, which is unnecessary. A detailed description should appear in the main text.
Reduced the framework outline and now rely more on the text.
- The quality of all figures and tables is poor. And the notes below the table should be concise.
Images were exported in high resolution tiff format and zipped with the text. Screenshots were just done to put images into text.
Notes below the tables have been reduced.
- ‘Mask’ is generally considered to be removed, for example, ‘water mask’ means to remove water bodies, so ' snow mask ' means to remove snow, but what you should want to express is snow cover, so ' snow masks ' is not suitable.
Yes the term “mask” can be confusing. I have clarified that I created analysis masks for the research and not cartographic masks that are used to mask out data.
MASK = [ESRI software] a means of identifying areas to be included in analysis. Such a mask is often referred to as an analysis mask, and may be either a raster or feature layer.
https://support.esri.com/en-us/gis-dictionary/mask
- Lines 160-166, the expression is not clear. Lines 160-162 say ‘When areas with potentially 100% snow cover are obscured with partial cloud cover (less than 100%), the cloud cover layer is added to the snow cover layer and the pixel value result equals 100%’. How to understand ‘potentially 100% snow cover’,’ partial cloud cover (less than 100%)’ and ‘pixel value result equals 100%’.
I have rewritten this section and hopefully it is clearer.
- The description of MOD10C2 data should be based on the ‘MODIS/Terra Snow Cover 8-Day L3 Global 0.05Deg CMG, Version 61’ user guide, and this reference may be more suitable than ‘MODIS snow products user guide for Collection [38]’.
Yes I have clarified this reference
- In Section 2 Materials and Methods, title 2.4 Regional Analysis is not suitable, because it is not a method.
Have removed section 2.4.
- According to the description in the text, this study should evaluate six major regions of the world with snow cover (Asia, North America, Europe, South America, Australia-New Zealand, and Africa) rather than five.
Yes this was a mistake – you are correct there are 6 regions.
- References 18, 22, 94 need to pay attention to formatting
Have done so.
Thank you for the helpful comment, they have greatly improved the manuscript.
Round 2
Reviewer 1 Report
All of my previous suggestions have been revised . I have no other comments. One small problem is that there is too much text in Figure 1, and the name of Figure 2 is missing.